# Melanization of *Candida auris* Is Associated with Alteration of Extracellular pH

**DOI:** 10.3390/jof8101068

**Published:** 2022-10-11

**Authors:** Daniel F. Q. Smith, Nathan J. Mudrak, Daniel Zamith-Miranda, Leandro Honorato, Leonardo Nimrichter, Christine Chrissian, Barbara Smith, Gary Gerfen, Ruth E. Stark, Joshua D. Nosanchuk, Arturo Casadevall

**Affiliations:** 1W. Harry Feinstone Department of Molecular Microbiology and Immunology, The Johns Hopkins Bloomberg School of Public Health, Baltimore, MD 21205, USA; 2Krieger School of Arts & Science, Johns Hopkins University, Baltimore, MD 21218, USA; 3Department of Microbiology and Immunology, Albert Einstein College of Medicine, Bronx, New York, NY 10461, USA; 4Division of Infectious Diseases, Department of Medicine, Albert Einstein College of Medicine, Bronx, New York, NY 10461, USA; 5Laboratório de Glicobiologia de Eucariotos, Departamento de Microbiologia Geral, Instituto de Microbiologia, Universidade Federal do Rio de Janeiro, Rio de Janeiro 21941-901, Brazil; 6Department of Chemistry and Biochemistry, City College of New York and CUNY Institute for Macromolecular Assemblies, The City University of New York, New York, NY 10031, USA; 7Institute for Basic Biomedical Sciences Microscope Facility, The Johns Hopkins School of Medicine, Baltimore, MD 21205, USA; 8Department of Biochemistry, Albert Einstein College of Medicine, Bronx, New York, NY 10461, USA

**Keywords:** melanin, *Candida auris*, alkalinization, ammonia, hydrophobicity, virulence

## Abstract

*Candida auris* is a recently emerged global fungal pathogen, which causes life-threatening infections, often in healthcare settings. *C. auris* infections are worrisome because the fungus is often resistant to multiple antifungal drug classes. Furthermore, *C. auris* forms durable and difficult to remove biofilms. Due to the relatively recent, resilient, and resistant nature of *C. auris*, we investigated whether it produces the common fungal virulence factor melanin. Melanin is a black-brown pigment typically produced following enzymatic oxidation of aromatic precursors, which promotes fungal virulence through oxidative stress resistance, mammalian immune response evasion, and antifungal peptide and pharmaceutical inactivation. We found that certain strains of *C. auris* oxidized L-DOPA and catecholamines into melanin. Melanization occurred extracellularly in a process mediated by alkalinization of the extracellular environment, resulting in granule-like structures that adhere to the fungus’ external surface. *C. auris* had relatively high cell surface hydrophobicity, but there was no correlation between hydrophobicity and melanization. Melanin protected the fungus from oxidative damage, but we did not observe a protective role during infection of macrophages or *Galleria mellonella* larvae. In summary, *C. auris* alkalinizes the extracellular medium, which promotes the non-enzymatic oxidation of L-DOPA to melanin that attaches to its surface, thus illustrating a novel mechanism for fungal melanization.

## 1. Introduction

*Candida auris* is an emerging fungal pathogen that is believed to have originated in marine wetlands and marshes [1,2]. *C. auris* was first described in a clinical setting in 2009, but patient samples as early as 1996 have been retrospectively determined to be *C. auris* [3,4,5]. Since its discovery, *C. auris* has caused outbreaks in at least 47 countries [3]. *C. auris* is often acquired in healthcare settings and is believed to cause infections via biofilms formed on medical equipment such as intravenous catheters, leading to bloodstream infection (candidemia) and dissemination to organs [6]. The emergence of *C. auris* as a human fungal pathogen is particularly concerning due to its remarkably high resistance to known antifungal therapies. This pathogen displays especially high resistance to azole and echinocandin classes of antifungal drugs [3,7], posing a major concern for clinicians treating fungal infections, since many of the commonly used and most efficacious antifungal therapeutics belong to these two drug classes. Hence, a deeper understanding of the physical properties of *C. auris* and how they contribute to virulence is urgently needed.

One unexplored aspect of *C. auris* physiology is whether it can produce the virulence factor melanin. Other *Candida* species, such as *C. albicans* and *C. glabrata,* produce this virulence factor [8,9,10,11]. Melanin is a black-brown, insoluble, acid-resistant pigment found throughout a plethora of life forms, and it has particular importance in the context of fungal virulence in mammals [12,13]. In fungi, melanin is typically produced through either the oxidation of catecholamines (DOPA melanin), the Tyrosine degradation pathway (pyomelanin), or polyketide synthase pathways (DHN melanin) [12]. Genes controlling melanization are affected by temperature, nutrient stress, and copper/metal ion concentration [14,15,16]; the biochemical processes can also be altered by the environmental antioxidant balance and pH [17,18]. Melanin is formed either within the cell in organelles termed “melanosomes” and then exported to the cell wall, or in the cell wall itself [19,20]. Once located within the cell wall, the pigment can be anchored through interactions with chitin, chitosan, and other cell wall components [9,21,22,23]. In some melanizing fungi such as *C. neoformans,* melanin interacts strongly with polysaccharides, lipids, and proteins, many of which are still intact following boiling in acid, lipid extraction, and enzymatic degradation. These fungi are thought to use such molecules as a scaffold for melanin deposition in the cell wall [19,24,25]. This melanin can also be released from the cell wall, as seen in *C. neoformans,* where melanin granules are shed into the extracellular space during cell wall remodeling and can be recovered from the supernatant [19,26].

Functionally, fungal melanin pigments have strong antioxidant properties that allow them to resist oxidative damage caused by the host immune cells, such as macrophage and neutrophil oxidative bursts [27]. Additionally, melanin can bind and inactivate antimicrobial peptides and antimicrobial enzymes that the host typically uses to degrade and kill fungi during infection, as well as antifungal drugs used to treat infections [28,29,30]. Fungal melanins located in the cell wall can alter cell wall composition and physically mask pathogen-associated molecular patterns (PAMPs) that would otherwise be recognized and bound by pathogen recognition receptors (PRRs). These changes may lead to diminished recognition by host immune cells. Conversely, one subtype of melanin—DHN melanin—can be recognized by the melanin sensing C-type lectin (MelLec) receptor, which is part of the human innate immune response and can enhance fungal clearance [31]. Melanin could also be presented to lymphocytes in a stimulatory context, which results in melanin-specific antibody production, which can inevitably lead to fungal opsonization, phagocytosis, and complement deposition [28,32,33].

In this study, we investigated the ability of 18 *C. auris* strains to produce melanin. The presence of melanin was confirmed through detection of a stable free-radical structure via electron paramagnetic resonance spectroscopy, a characteristic feature of this group of pigments. Like other fungi, we found that *C. auris* produces a black melanin pigment, which was located on the surface of the cell wall. However, we found that, unlike other fungi, the melanization occurs extracellularly, in the supernatant. In liquid culture, the cell-free melanin can adhere to the surface of the cell and cause pigmentation of the fungus. Next, we found this process to be mediated by the ability of *C. auris* to neutralize the pH of the media it is grown in, resulting in enhanced autoxidation of the L-DOPA and catecholamine melanin substrates. This method of melanin production differs greatly from what is reported in other fungi. Lastly, we evaluated the functional properties of *C. auris* melanin, finding that it protected the fungus from oxidative stress and reduced cell surface hydrophobicity but was not protective in vitro in the context of interactions with macrophages, or in an in vivo challenge of *Galleria mellonella* larvae. Further studies are required to identify the unique features of *C. auris* that make it particularly resistant to antifungal therapeutics.

## 2. Materials and Methods

### 2.1. Candida auris Strains and Media

All isolates of *Candida auris* strains were received from the Centers of Disease Control and Prevention Food and Drug Administration Antimicrobial Resistance Isolate Bank [34] with the exceptions of MMC1 and MMC2, which were previously described clinical isolates [35]. CDC 381 is also known as B11220, CDC 387 is also known as B8441, and CDC 388 is also known as B11098. All *C. auris* strains were first grown in Yeast Peptone Dextrose (YPD) broth at 30 °C until they reached stationary phase. Cultures were washed twice and put into Minimal Media (15.0 mM glucose, 10.0 mM MgSO_4_, 29.4 mM KH_2_PO_4_, 13.0 mM glycine, 3.0 M vitamin B_1_, pH 5.5) at 37 °C at 10^7^ cells/mL unless otherwise noted. Cells were grown for 7 days under continuous shaking.

### 2.2. C. auris Melanization in Liquid Media

All *C. auris* strains were grown in minimal media as described above, with the addition of melanin precursors to be tested. All compounds used for the melanization substrate assay and ABTS (2,2′-azino-bis(3-ethylbenzothiazoline-6-sulfonic acid) laccase assay were prepared in concentrated stock solutions and added to the culture at a final concentration of 1 mM except for caspofungin, which was added at 0.5 µg/mL final concentration. For the experiments evaluating the temperature dependence of the *C. auris* strain melanization, cultures were also grown at 30 °C for 7 days in minimal media. After 7 d, cultures were scanned with a CanoScan9000F scanner at 600 dpi. Mean Gray Value for each well of culture was determined using the Measure tool on FIJI image processing software [36].

For experiments where cultures were grown on solid agar petri dishes, 20 µL of stationary washed culture were added to minimal media agar with 1 mM of L-3,4-dihydroxyphenylalanine (L-DOPA). Plates were incubated at 37 °C for 7 days and imaged with a CanoScan9000F scanner at 600 dpi.

### 2.3. Melanin Extraction and Electron Paramagnetic Resonance (EPR)

Stable free-radicals, a hallmark feature of melanin, can be detected through electron paramagnetic resonance (EPR). The ability to detect stable free-radicals lends to EPR’s usefulness as a technique to characterize and identify melanin pigments. Melanin was extracted from selected strains of *C. auris* and one strain of *C. neoformans* (H99) as described [37]. Briefly, after growing for 7 days in minimal medium containing L-DOPA (1 mM), cells were washed with sorbitol/sodium citrate solution and incubated with Novozyme 234 for 1 h at 30 °C. Samples were incubated with guanidine thiocyanate for 1 h, followed by an incubation with 6 M HCl. After boiling the solution for 1 h, the pellets were washed and suspended in PBS. Extracted melanin was examined using a Varian E112 X-Band model spectrometer with a TE102 resonator and a liquid nitrogen finger Dewar vessel to obtain EPR spectra of the collected dark particles suspended in PBS and frozen with liquid nitrogen. The EPR runs were performed with the following parameters that were standardized for our fungal melanin studies as described previously [8]: modulation frequency of 9.07 GHz, modulation amplitude of 1.6 G, center field of 3250.0 G, sweep width of 100.0 G, microwave frequency of 9.1 GHz, microwave power of 1.0 mW, time constant of 0.5 s, and temperature of 77 K.

### 2.4. Extracellular Melanin Isolation

Cultures grown for 7 days with or without L-DOPA as a substrate for melanization were centrifuged at 4000× *g* for 5 min. The supernatant was removed and sterilized through a 0.8 µm syringe filter (Corning, Corning, NY, USA). To measure the melanization of the supernatant, absorbance measurements were performed at 492 nm using a SpectraMax iD5 spectrophotometer. For experiments collecting and analyzing the extracellular melanin particles, the supernatants were then ultracentrifuged at 100,000× *g* for 1 h at 4 °C in a Beckman Coulter Optima L-90K UltraCentrifuge. Supernatants were decanted, and the melanized pellet was suspended in PBS.

### 2.5. Supernatant Melanization Activity

To assess phenoloxidase activity of the supernatant, L-DOPA was added to cell-free supernatant from non-melanized cultures. Subsequently, the supernatant was left to incubate for 72 h at 37 °C in darkness. Activity was determined through SpectraMax iD5 spectrophotometer readings at 492 nm taken at 0 and 72 h, and images were taken using a CanoScan9000F flatbed scanner at a resolution of 600 dpi. For the proteinase assays, supernatants were pre-treated with 1:10 Trypsin (Corning) at 37 °C, or with approximately 200 µg/mL Proteinase K (New England BioLabs) at 60 °C prior to the addition of L-DOPA. To measure the degree of protection conferred by extracellular vesicles against degradation of potential melanin-producing components, supernatants were pre-treated with 0.01% *w/v* SDS or 0.1% *v/v* Triton X-100 (Sigma-Aldrich, St. Louis, MO, USA) with or without Proteinase K. Supernatants were then incubated for one hour at 60 °C as described. Heat degradation assays were performed by heating supernatant samples to 100 °C for one hour prior to testing for melanization activity.

### 2.6. Light Microscopy

*C. auris* strains were imaged using light microscopy using an Olympus AX70 microscope and 100× objective.

### 2.7. Transmission Electron Microscopy (TEM)

*C. auris* melanized and non-melanized cells were imaged using TEM as described [26]. Briefly, samples were fixed with 2% (*w/v*) glutaraldehyde in 0.1 M cacodylate at room temperature for 2 h, followed by overnight incubation in 4% (*w/v*) formaldehyde, 1% (*w/v*) glutaraldehyde, and 0.1% PBS overnight at 4 °C. Samples were washed, fixed with 1% osmium tetroxide for 90 min, washed with dH_2_O, serially dehydrated in ethanol, and embedded in SPURRS resin. Thin sections, 60 to 90 nm, were cut with a diamond knife on a Leica Ultracut E Ultramicrotome and picked up with 2 × 1 mm formvar-coated copper slot grids. Grids were stained with 2% uranyl acetate (aq) and 0.4% lead citrate before imaging on a Hitachi 7600 TEM at 80 kV. Images were captured with an AMT XR80 CCD (8 megapixels, 16 bit). Secreted melanin was visualized using negative staining, in which 8 µL of sample was placed on negative glow discharged 400 mesh ultra-thin carbon-coated grids (EMS CF400-CU-UL) for 30 s, followed by three quick rinses of Tris-buffered Saline (TBS) and staining with 2.5% uranyl acetate. Samples were imaged using a Hitachi 7600 TEM Electron Microscope at 80 kV. Images were captured with an AMT XR80 CCD (8 megapixels, 16 bit). Following image acquisition, diameter measurements of the secreted melanin particles were performed using the measurement tool of FIJI image processing software [36].

### 2.8. Scanning Electron Microscopy (SEM)

Briefly, samples were fixed in 2.5% (*v/v*) glutaraldehyde in 0.1 M sodium phosphate buffer (pH 7.3) overnight at 4 °C. Samples were placed on a poly-L-lysine-coated coverslip (0.01 mg/mL, coated for 5 min, and rinsed twice in dH_2_O) for 1 h, then washed, serially dehydrated in ethanol, chemically dried using hexamethyldisilazane (HMDS) and dried overnight in a desiccator. Samples were then placed on a metal sample stub (EMS aluminum 6 mm pin, 12.7 mm diameter) with double-sided carbon tape (EMS standard carbon adhesive tabs, 12 mm diameter) and the underside of the coverslip coated with silver paint (EMS silver conductive coating). Samples were coated with 15 nm gold palladium (AuPd), on a Denton Desk III sputter coater before imaging on a ThermoFisher Helios FIB-SEM at 5 kV using an Ion Conversion and Electron (ICE) detector.

### 2.9. Extracellular Melanin Add-Back

Cell-free melanized supernatant was added to a pellet of non-melanized cells from the corresponding strain of *C. auris.* Samples were mixed at 37 °C for 3 h and pelleted at 4000× *g* for 5 min. Pelleted cells were imaged compared to control cells treated for 3 h with the non-melanized supernatant.

### 2.10. Cell Wall Disruption Assays

All cultures were grown in minimal media for 7 days with 1 mM L-DOPA. In addition to the L-DOPA, either 5 mM N-acetylglucosamine (GlcNAc) as previously described [9], 100 µg/mL Calcofluor White as previously described [9], or 50% MIC of Caspofungin based on the CDC resistance profile [34] (Antibiotic Resistance Isolate Bank, Centers for Disease Control and Prevention, Atlanta, GA, USA) were added to the culture at the beginning of the 7 d incubation. Cultures were collected after 7 d; the supernatant and cells were examined under light and electron microscopy as described above.

### 2.11. Preparation of Fungal Cells for ssNMR Analysis

*C. auris* cells from each of the CDC 387, CDC 388, and CDC 381 strains were grown with and without L-DOPA in separate flasks using the same culture conditions as described above. To verify the ability of *C. auris* to take up and utilize exogenous GlcNAc for chitin synthesis, an additional culture of CDC 388 cells was prepared in growth medium supplemented with 5 mM ^15^N-enriched GlcNAc. The cells from all cultures were harvested via centrifugation, and the resulting pellets were resuspended in 25 mL deionized water. To heat-kill the cells, the tubes containing these cell suspensions were immersed in a water bath at 65 °C for 1 h. When cooled to room temperature, the heat-killed cells were centrifuged at 3,700 rpm for 30 min at 4 °C. The pellets were resuspended in another 25 mL aliquot of deionized water, vortexed vigorously, and again centrifuged. This process was repeated four more times to remove any residual metabolites, cellular debris, or other small molecules. After the fifth wash, the cell pellets were lyophilized for 3 days and subsequently analyzed by ssNMR.

### 2.12. Solid-State NMR Spectroscopy

All measurements were carried out on a Varian (Agilent) DirectDrive2 (DD2) instrument operating at a ^1^H frequency of 600 MHz and equipped with a 1.6-mm T3 HXY fastMAS probe (Agilent Technologies, Santa Clara, CA). The data were acquired on 6–8 mg of lyophilized cell material using a MAS rate of 15.00 ± 0.02 kHz at a spectrometer-set temperature of 25 °C. The ^13^C direct-polarization magic-angle spinning (DPMAS) experiments were conducted with 90° pulse lengths of 1.2 and 1.4 μs for ^1^H and ^13^C, respectively; 104-kHz heteronuclear decoupling using the small phase incremental alternation pulse sequence (SPINAL) was applied during signal acquisition. Long recycle delays (50 s) were implemented to obtain spectra with quantitatively reliable signal intensities that allowed the integration of defined spectral regions using the GNU image manipulation program (GIMP) to estimate the relative amounts of carbon-containing constituents in heat-killed intact *C. auris* cell samples. To verify the uptake of ^15^N-enriched GlcNAc, ^15^N cross-polarization (CPMAS) experiments were conducted using pulse lengths of 1.6 and 2.9 μs for ^1^H and ^15^N, respectively, a 1.5-ms cross polarization period with a 10% linear ramp, and 78-kHz SPINAL decoupling during acquisition.

### 2.13. pH Measurements and Supernatant pH Alteration

Cultures were grown for 7 days in minimal media at 37 °C. Cultures were centrifuged at 4000× *g* for 4 min, and the supernatant was filter sterilized with a 0.22 µm PES Filter (Millipore-Sigma, St. Louis, MI, USA). The supernatant pH was determined using a calibrated Fisher Scientific Accumet AB150 pH meter.

Cell-free supernatants were isolated and their pH values measured using a calibrated Fisher Scientific Accumet AB150 pH meter. Samples were split into three groups: unaltered supernatant, supernatant manually adjusted to pH 5.5 with hydrochloric acid (HCl), and supernatant manually adjusted to pH 7 with potassium hydroxide (KOH). Titrations with HCl and KOH were performed using the same pH meter set to continuously read pH. Each sample’s three groups were then treated with 1 mM L-DOPA and incubated at 37 °C for 72 h in darkness. Melanization activity was determined through SpectraMax iD5 spectrophotometer readings at 492 nm, and images were taken using the CanoScan9000F flatbed scanner at a resolution of 600 dpi.

### 2.14. Ammonia Quantification

Supernatant ammonia concentration was quantified using the commercially available API Ammonia Test Kit (API) according to the manufacturer’s protocol, using the Solutions 1 and 2 from the kit. The sample volumes were scaled down in proportion to the small volumes tested to maintain the established test solution concentrations. A standard curve of ammonia concentrations from 16 ppm to 0.25 ppm was constructed by serially diluting 28% ammonium hydroxide (Sigma) in PBS to enable validation of this modification and colorimetric correlation to known ppm values. In a 48-well plate, 1 drop of both Solution 1 (>60% *w/v* polyethylene glycol, 1–10% *w/v* sodium nitroprusside solution, 1–10% *w/v* sodium salicylate) and Solution 2 (<10% *w/v* sodium hydroxide, <1% sodium hypochlorite) were added to 625 µL of cell-free supernatant, after which the plate was agitated and left to sit for 5 min with a lid on. The color change, from yellow to dark blue-green during the formation of indophenol blue dye, was quantified by reading the absorbance at 680 nm using a SpectraMax iD5 spectrophotometer. In addition, plates were imaged at 600 dpi using a CanoScan9000F flatbed scanner.

### 2.15. Cell Surface Hydrophobicity (CSH)

Cell surface hydrophobicity was measured by Microbial Adhesion to Hexadecane (MATH) assay as described [38]. Briefly, cells were resuspended in PBS to an optical density of 0.2–0.4 at 600 nm using a SpectraMax iD5 spectrophotometer. These values were measured in triplicate and recorded as the initial optical density. In total, 3 mL of cells in PBS were added to a glass test tube followed by 400 µL of n-hexadecane. Tubes were covered with parafilm and vortexed on high for 45 s each and left to settle for 2 min. Aliquots of the aqueous (bottom) layer were carefully removed, and the absorbance at 600 nm was measured as the final optical density value. Hydrophobicity was calculated as: 100 × (Initial Value − Final Value)/(Initial Value).

### 2.16. Oxidative Stress

Yeast cells grown in the presence or absence of L-DOPA for 7 days at 37 °C were incubated and shaken in RPMI medium buffered with MOPS, with or without 5 mM hydrogen peroxide (H_2_O_2_) for 3 h at 37 °C. Cell suspensions were diluted and plated onto Sabouraud-agar plates for colony-forming units (CFU) counting. The cytotoxic effect of H_2_O_2_ in each strain was calculated as the number of yeast cells in the presence of H_2_O_2_, divided by the yeast count in the absence of H_2_O_2_, multiplied by 100 to be expressed as a percentage.

### 2.17. Killing by Bone Marrow-Derived Macrophages

Bone marrow cells isolated from C57BL/6 mice were cultivated in RPMI containing 10% of FBS and 20% of L929 supernatant for 7 days with media addition on day 3. At the end of the differentiation time, cells were plated in 96-well plates (10^5^ cells per well) and incubated at 37 °C to achieve adherence. Macrophages were challenged with *C. auris* (0.5 × 10^5^ yeast cells per well) that were grown in the presence or absence of 1 mM L-DOPA. In parallel, *C. auris* was added to wells under the same conditions but without macrophages. After 2 or 24 h, the plates were centrifuged, the supernatant discarded, and the pellets suspended in sterile distilled H_2_O. The suspensions were diluted and plated onto Sabouraud-agar plates and incubated at 30 °C for 24 h for CFU counting. For each experimental group, the percentage of yeast killing was calculated as: yeast + macrophage group divided by yeast in the absence of macrophages.

### 2.18. Galleria Mellonella Infection

Groups of 10 insects (250–300 mg) in the final instar larval stage were used. Larvae were injected with melanized or non-melanized yeasts of *C. auris* (2 × 10^6^ cells in 10 µL) into the haemocoel through the last left pro-leg strains using a Hamilton syringe. For these experiments, the strains CDC 381, CDC 387, and CDC 388 were selected. PBS was used as a control. The insects were then placed in sterile Petri dishes and maintained in the dark at 37 °C. The numbers of living larvae were monitored twice daily and recorded for a period of 7 days. Larvae were considered dead if no response to physical stimulus was observed.

## 3. Results

### 3.1. Temperature Dependence of Melanization on Candida auris Strains

To evaluate the ideal temperature for *Candida auris* melanization, we tested the degree of melanization at both 30 °C and 37 °C of several strains across five *C. auris* clades. We found that all strains melanized to a greater extent at the higher temperature compared to the lower temperature (Figure 1A–C). The melanin-capable strains of *C. auris* were CDC 385, CDC 386, CDC 387, CDC 388, CDC 389, CDC 390, CDC 931, CDC 1097, CDC 1104, and MMC1 (Table 1). Strain CDC 382 exhibited an intermediate melanin phenotype. Additionally, we note that certain strains, namely CDC 388, CDC 390, CDC 1097, and MMC1, melanized significantly more at 37 °C and not at the lower temperature. This trend has interesting implications for understanding regulation of melanization and the role it has in warm environmental niches, as well as during infections of humans, who have a normal core body temperature of ~37 °C.

Further, the capacity of individual strains to melanize was associated with their evolutionary clade, with strains belonging to Clades I and IV—associated with South Asia and South America, respectively—demonstrating melanization activity, while those strains belonging to Clades II and III—associated with East Asia and Africa, respectively—did not (Table 1). In addition, the single representative of Clade V—associated with Iran—melanized, but the lack of additional Clade V isolates for testing hinders the generalization of this finding. The antifungal drug susceptibility (Minimum Inhibitory Concentrations) of the melanin-producing strains as a whole was generally higher than the susceptibility of non-melanin-producing strains as a whole (Appendix A), as reported by the CDC and FDA Antibiotic Resistance Isolate Bank [34]. This association may be related to the ability to melanize or the previously reported clade-specific genetic or physiologic differences, including those related to drug susceptibility [39,40].

### 3.2. Electron Paramagnetic Resonance (EPR)

Melanin is characterized by a stable free-radical structure, which lends to its antioxidant properties. Stable free radicals can be detected by electron paramagnetic resonance (EPR), making this biophysical technique the ‘gold standard’ for the identification of melanin pigments. Melanin from 7-day cultures was extracted from 12 strains of *C. auris* and analyzed by EPR, where melanin extracted from *C. neoformans* H99 was used as a standard (Figure 1D–F). The EPR spectra from the non-melanizing strains did not display the melanin-distinctive EPR peak (Figure 1E), but the EPR spectra from the pigment extracted from the melanin-producing strains (Figure 1F) was similar to the *C. neoformans* melanin profile (Figure 1D).

### 3.3. Time and Cell Density

To understand the optimal conditions for melanin production by *C. auris*, yeast cells were incubated with L-DOPA starting with different cell densities, and the production of pigment was analyzed every 2 to 3 days over a 13-day period and compared to cells incubated in the absence of L-DOPA (Figure 2). *C. neoformans* was also cultured under the same conditions and compared to the *C. auris* melanization profile. Melanin production was optimal by cells in high density (10^7^ cells/mL). In this condition, melanin was visible from day 3, and intensified over a period of days, reaching its peak between days 7 and 10. Some strains melanized in a delayed fashion when grown at a medium cell density (10^6^ cells/mL), whereas none of the low-density (10^5^ cell/mL) cultures produced visible pigment. The cell density-associated phenotype presented by *C. neoformans* was different from that presented by *C. auris*, as melanin production by low cell density *C. neoformans* cells was equally or more effective than in the high-density conditions.

### 3.4. Candida auris Strains Melanize Using Substrates Associated with DOPA Melanin

We examined various known melanin precursors to evaluate which of them the *C. auris* strains could use as substrates for melanization, and if there are any strain-specific differences between the substrates used. We found that at a starting inoculum of 10^7^ cells/mL, melanin-capable *C. auris* strains were able to use L-3,4-dihydroxyphenylalanine (L-DOPA), D-3,4-dihydroxyphenylalanine (D-DOPA), L-methyl-3,4-dihydroxyphenylalanine (Methyl-DOPA), dopamine, norepinephrine, epinephrine, and a mixture of catecholamine neurotransmitters (dopamine, epinephrine, and norepinephrine; ‘brain mix’) at the ratios found in the mammalian brain [41]. We found that in general, all melanin-capable *C. auris* strains were able to use the same substrates (Figure 3A). Some strains, namely CDC 385 and CDC 386, were less able to produce pigment when grown with dopamine. Whereas the L-DOPA, D-DOPA, Methyl-DOPA, dopamine, brain mix, and norepinephrine resulted in the formation of pigments that were dark brown to black in color, the epinephrine precursor resulted in reddish-brown or amber colored melanization (Figure 3B).

Notably, all the *C. auris* strains tested were unable to use L-Tyrosine, homogentisic acid (HGA), or 4-hydroxyphenylpyruvic acid (4-HPP) as precursors for melanization. First, this indicates that *C. auris* does not have a tyrosinase enzyme that can convert L-Tyrosine into a diphenolic precursor suitable for DOPA melanin synthesis. Second, this indicates that *C. auris* cannot produce pyomelanin, which is synthesized from intermediate products generated along the Tyrosine Degradation Pathway (in which 4-HPP and HGA participate) and typically involves laccase-mediated polymerization [42]. Additionally, the strains were also unable to oxidize ABTS, a laccase-specific substrate, further indicating that the melanization agent is not a laccase (Figure 3C).

### 3.5. Melanin Is Primarily Found in the Supernatant

During collection of the melanized cells, we noted that the supernatants of the cultures were notably darker than expected given our work with another melanizing fungus, *C. neoformans* (Figure 4A,B). Similarly, when cells were grown on agar plates with L-DOPA as a substrate for melanization, there was a distinct halo of pigment surrounding the melanin-producing strains of *C. auris (*Figure 4C). Notably, there was little to no pigmentation of the yeast colony itself. This observation strongly suggested that the melanization of the cells was primarily extracellular and did not originate within the cells themselves.

To evaluate whether the supernatant of the cells had melanin-producing enzymatic activity, we added L-DOPA to the isolated supernatant. We found that the supernatants of melanizing strains, namely the supernatant of MMC1, CDC 387 (B8441), and CDC 388 (B11098), were capable of oxidizing L-DOPA, whereas the supernatant of the non-melanizing strain CDC 381 (B11220) and minimal media alone were not. This capability supported the hypothesis that the melanin-producing components of the *C. auris* were secreted (Figure 4D).

Using ultracentrifugation, we collected small particles and extracellular vesicles found within the melanized supernatant of the liquid cultures. We found that a substantial amount of the pigment was in this supernatant pellet. Using negative staining transmission electron microscopy (TEM), we found that the pelleted melanin was organized within granule structures, comparable to what was seen at the cell wall periphery of the melanized culture (Figure 4E,F). These structures measured 20–40 nm (Figure 4E) and were similar in appearance to melanin granules secreted by *C. neoformans* [19]. Figure 4F illustrates that the granules from CDC 381 had the smallest mean diameter (~20 nm) and CDC 387 had the largest value (~40 nm).

### 3.6. Cell-Bound Melanin Is Localized to the Periphery of the Cell Wall

To visualize the localization of melanin on the *C. auris* cells grown in liquid media, we used light microscopy, TEM, and scanning electron microscopy (SEM). Under the light microscope, dark pigmentation of the cells appeared to be located primarily within or proximal to the cell wall. Some cells were observed to have additional intracellular pigmentation, possibly due to oxidation of L-DOPA within a large vacuole (Figure 5A). Comparing TEM micrographs of melanized and non-melanized cultures, we found electron-dense structures exclusively in the melanized cultures, primarily on the periphery of the cell wall. These electron-dense structures are likely to be melanin that appeared to be rounded and granular. The extracellular melanin granules were also unbound to the cell wall (Figure 5B). These extracellular melanin granules were similar to the secreted melanin granules previously reported in *C. neoformans* [19] and were roughly 20–40 nm in diameter (Figure 4E,F and Figure 5B). Similarly, samples imaged with SEM showed that melanized cells had raised structures on their surface, consistent with the structures observed by TEM (Figure 5C). Further, in the case of CDC 387, 388, and MMC1, the melanin granules appeared to hold the cells together in large, aggregated clumps.

To determine the directionality of the cell-bound melanin deposition (i.e., if melanin was formed in the cell wall and then released into the media, or if the pigment was formed primarily in the media and then attached to the cell-wall periphery), we performed a series of ‘add-back’ experiments, in which we added melanized supernatant to the cells from the same *C. auris* strain grown without L-DOPA. We found that the cells became pigmented by three hours of incubation with the melanized supernatant (Figure 6A–D). Additionally, when we added melanized supernatant from the CDC 388 strain to the non-melanizing CDC 381 strain, the CDC 381 strain accumulated pigment (Figure 6D). First, since CDC 381 cannot readily produce pigment, this experiment showed that the pigment accumulation in these add-back experiments was due to adhesion of already synthesized pigment in the supernatant rather than the production of new pigment from unreacted L-DOPA in the supernatant. Second, it indicated that the melanin-deficient cells did not have a cell wall difference that made them intrinsically unable to bind extracellular melanin. These add-back experiment data, along with the secretion data of Figure 4 and Figure 5, point to the conclusion that *C. auris* cell wall melanization occurs extracellularly and then sticks to the outside of the cell wall.

### 3.7. Effects of Altering Cell Wall Structures on the Melanization of C. auris

To determine what cell wall components were important for adherence of the melanin to the *C. auris* cell wall, we grew CDC 387 and CDC 388 with L-DOPA in the presence of compounds known to enhance or block the proper formation of cell-wall components [9,43], either through direct supplementation of these cell wall components that bind and block proper structure formation, or by inhibition of the enzyme responsible for producing that cell wall component. This strategy allowed us, in theory, to evaluate which components of the cell wall were important for extracellular melanin adherence. Culture conditions that resulted in a darker supernatant than the control condition indicated the lack of melanin adherence to the cell wall, and those with lighter supernatant indicated that the compound enhanced melanin adherence to the cell wall.

The cell-wall polysaccharide chitin, a polymer of β-1,4-linked N-acetylglucosamine (GlcNAc) units, has been demonstrated to play a role in melanization in *C. neoformans* and *C. albicans* [9,44]. In these organisms, supplemental GlcNAc provided in the cell-culture media was used as a substrate for chitin formation. This in turn increased the overall content of chitin in the cell wall, which in *C. albicans* results in increased melanin production, externalization, and cell-wall adhesion, and in *C. neoformans* results in increased cell-wall melanin deposition and retention. To determine whether a similar relationship exists between chitin synthesis and melanization in *C. auris*, cells grown in culture medium supplemented with GlcNAc that was enriched in the NMR-active ^15^N-isotope were examined using solid-state NMR spectroscopy (ssNMR) to determine the metabolic fate of the labeled exogenous substrate. The observation of an ^15^N NMR signal corresponding to the amide nitrogen of chitin verified that *C. auris* is indeed capable of taking up exogenously provided GlcNAc and subsequently using it as a precursor for chitin synthesis (Appendix A). In CDC 387 and CDC 388, addition of 5 mM GlcNAc did not enhance cell wall melanin adhesion and instead resulted in darker supernatant, perhaps indicating less adhesion (Figure 6E,F). In TEM micrographs, the melanin located at the cell wall exterior did not appear appreciably different for the control and the GlcNAc treated cultures (Figure 6G,H).

Conversely, we added 100 µM Calcofluor White (CFW), a fluorescent dye which forms hydrogen bonds with chitin polymers as they grow and thus disrupts proper formation of chitin microfibrils [45,46]. Adding this dye led to markedly decreased supernatant pigmentation (Figure 6E,F). Intriguingly, under microscopic analysis, the CFW appeared to precipitate and form crystals that bound the melanin. Thus, the adhesion of melanin to the CFW crystals was the likely cause of the lack of pigmented supernatant. This resulted in black crystals that had the fluorescent properties of the CFW, along with the expected CFW-stained cell walls (Appendix A). The CFW crystals were also visible under electron microscopy (Figure 6H, Appendix A). The crystallization persisted despite filter sterilization. In an assay of cell-free supernatant with CFW and with/without L-DOPA, we confirmed that the synthesized melanin bound to the CFW crystals and precipitated out of solution (Appendix A). We also noticed that the CDC 387 and CDC 388 strain supernatants without L-DOPA formed a smaller population of CFW crystals compared to the CDC 381 non-melanizing strains (Appendix A). This finding may indicate a correlation between the factor responsible for melanization and formation of CFW crystal fragments with a smaller size, which could indicate better solubility.

Further, we grew the CDC 387 and CDC 388 strains in the presence of the antifungal compound caspofungin at half of the concentration reported by the Centers for Disease Control and Prevention (CDC) [34] to inhibit the enzymes responsible for cell wall β-glucan synthesis. Caspofungin treatment resulted in high levels of supernatant melanization (Figure 6E,F). Upon light microscopic and electron microscopic analysis, the treatment appeared to reduce the amount of melanin present on the cell wall, with nearly no melanin visible on the cell wall via TEM (Figure 6G,H). Additionally, for the CDC 388 strain, caspofungin treatment resulted in what appeared to be pseudohyphal growth (Appendix A). Although the cells were able to grow in culture with this concentration of caspofungin, in the electron micrographs, the cells appeared to have a deformed morphology, and their cytoplasmic contents looked condensed and abnormal, potentially due to antifungal stress and cell-wall defects (Figure 6G,H).

### 3.8. Melanization Affects the Hydrophobicity of Some C. auris Strains

Since melanin is a hydrophobic molecule, we investigated whether melanization had a correlation with the cell surface hydrophobicity (CSH) of the *C. auris* strains. With *C. auris* strains grown in the nutrient-rich Yeast Peptone Dextrose (YPD) media and minimal media, 7 of 12 strains had CSH that was greater than 80% (Figure 6I,J). There was no correlation between the strain hydrophobicity and its ability to melanize; the strongly melanizing strain CDC 387 and the weak/no melanizing strains CDC 381 and MMC2 all had low hydrophobicities. Conversely, the non-melanizing strain CDC 383 and melanizing strain CDC 388 had high hydrophobicity. Additionally, since hydrophobicity is known to play roles in host-pathogen interactions [47,48,49], we investigated whether melanization affected the CSH of the *C. auris* strains. Surprisingly, we found that in the CDC 387 strain (and to a lesser extent in the non-melanizing strain CDC 381), growth in L-DOPA for 7 days resulted in decreased CSH compared to the controls grown without L-DOPA (Figure 6K). In CDC 388 and MMC1, we did not see a change in CSH following melanization, although these strains had notably high levels of hydrophobicity at ~90%.

### 3.9. Melanin Adherence Is Correlated with a Higher Cell-Wall Polysaccharide Content

To further assess a potential role for the cell wall in *C. auris* melanization, we used ^13^C solid-state NMR (ssNMR) to determine whether the proportion of cell-wall polysaccharides in these cell samples differs between the two melanizing *C. auris* strains evaluated in this study, CDC 387 and CDC 388, and with respect to the non-melanizing strain CDC 381. To circumvent the need to hydrolyze and purify the cell-wall material for quantitation, we took the more direct approach of conducting direct-polarization magic-angle spinning (DPMAS) experiments on intact heat-killed *C. auris* cells. The experiments were carried out using parameters optimized to generate quantitatively reliable spectra, in which the integrated peak intensities represent the relative amounts of the different carbon types present in the various samples (Figure 7A). Thus, the relative number of polysaccharides present in each *C. auris* whole-cell sample could be estimated by measuring the integrated signal intensity within the spectral region where polysaccharide carbons resonate (~54–110 ppm) and comparing it to the total integrated signal intensity across the spectrum. To determine whether the process of melanization elicits changes in cell-wall composition, we examined cells from the three strains, each grown in the presence and absence of L-DOPA. Since melanin pigments have been reported previously to comprise only a minor fraction (1–15.4%) of the total dry mass of melanizing fungal cells [50,51] and are likely to have large molar masses, the NMR signals that arise from pigment carbons make a negligible contribution to the overall spectral intensity. Consequently, we were able to make quantitative estimates of relative polysaccharide content regardless of whether melanin was present.

Our analysis revealed that cells from the CDC 387 strain, which produce melanin that attaches robustly to the exterior cell wall, display a moderate difference in relative polysaccharide content depending on whether the cells are grown with or without L-DOPA (59% vs. 67% for melanized and non-melanized cells, respectively) (Figure 7B). In contrast, the relative polysaccharide content in CDC 388, where melanin is primarily found in the supernatant, differed by only 2% between cells grown with or without L-DOPA (52% vs. 54%). Polysaccharide content was also constant at 52% for the non-melanizing CDC 381 strain in cultures grown with and without L-DOPA. Interestingly, the most significant difference in relative polysaccharide content emerges if we compare the three strains rather than focusing on whether L-DOPA is present in any of their growth media. The average relative polysaccharide content of the CDC 387 cells (with and without L-DOPA) is 63%; the corresponding quantities for CDC 388 and CDC 381 cells are 53% and 52%, respectively. Taken together, our data suggest that the cells from the CDC 387 strain have an overall greater relative polysaccharide content in comparison to CDC 388 and CDC 381 cells, regardless of whether L-DOPA is present in the culture medium. This trend could potentially account for the robust cell-wall melanin binding ability of the CDC 387 strain.

### 3.10. Supernatant Melanization Is Resistant to Denaturing Conditions

To further elucidate the identity of the melanin-producing components, we sought to characterize properties and constraints on supernatant melanization. We first assessed the supernatant’s melanization activity after exposure to high temperatures. We observed a modest reduction in the supernatant’s capacity to melanize after a 1 h incubation at 100 °C, and no significant difference between the melanization activity of the precipitate or supernatant that formed after boiling (Figure 8A). After treating samples with the serine proteinases Trypsin or Proteinase K, we assessed the supernatant melanization activity and observed no reduction in the capacity for melanization (Figure 8B–D). Further, melanization activity was unaffected by treatment with 1% *w/v* SDS or methanol; however, melanization was increased after treatment with 6 M urea under all conditions, indicating that urea affected the melanization reaction irrespective of the melanizing strain-specific factor in the supernatant (Figure 8E). These observations argue against the melanization factor being a protein.

We sought to characterize the size of the melanin-producing components. After passing the supernatant through a 3 kDa protein cut-off filter, we found that melanization activity was partially conserved in the flowthrough of the concentrator, suggesting that the melanization factor was not larger than 3 kDa (Figure 8F).

To check whether a melanin-producing component was being protected from denaturation within Extracellular Vesicles (EVs), we treated the supernatant with Triton X-100 and SDS in combination with proteinases. If the melanization factor were protected within EVs, adding detergent would likely compromise EV integrity and allow the protease to degrade the protein contents within [52,53,54]. We did not find that EVs protected a melanizing factor from proteolytic degradation (Figure 8G). This finding was further supported by the absence of enhanced melanization activity for an EV-enriched ultracentrifuged pellet (Appendix A).

### 3.11. Melanization Genes in the C. auris Genome

We searched the *C. auris* genome for enzymes that have been associated with melanin synthesis in other fungi such as laccases, phenol oxidases, and tyrosinases. We found a number of hypothetical multicopper oxidase and ferroxidase genes reported in the *C. auris* genome. However, we have no evidence that these have laccase or tyrosinase-like functions; they appear similar to metal ion transporters, peroxisomal membrane components, and ergosterol synthesis (Appendix A). Hence, the absence of tyrosinase and laccase activity in *C. auris* is in accord with the absence of a clearly identifiable enzyme in the genome.

### 3.12. Supernatant Neutral pH Correlates with Ability of C. auris Strains to Melanize

Due to the smaller crystal size of the CFW in the supernatants from the melanizing strains, we measured the pH of their supernatants, since alkaline pH can enhance the solubility of CFW solutions [55,56]. Additionally, the urea treatment, which produced an alkaline environment, increased supernatant melanization in all the strains. We found a strong correlation between the pH of the conditioned media and the degree to which the cultures melanized (Figure 9A). We found that the melanizing strains have a supernatant of about pH 6.5, whereas the non-melanizing strains tended to have a supernatant closer to pH 5.75. The higher values of supernatant pH fall in the range at which L-DOPA and other catecholamines auto-oxidize to melanin (Figure 9B).

We altered the supernatant pHs to obtain values of either pH 5.5 or 7 by adding hydrochloric acid or potassium hydroxide, respectively. We found that melanization activity was halted at pH 5.5, including the supernatants of CDC 387 and CDC 388, but was enhanced at pH 7 (Figure 9C,D). Most notably, the supernatant from CDC 381—a non-melanizing strain—achieved melanization activity at higher pH that was statistically indistinguishable from that of the melanizing strains—CDC 387 and CDC 388 at the same pH. This observation strongly suggests that the pH of the supernatant is a major contributor to the observed extracellular melanization activity.

As endogenous production of ammonia has been reported in *C. albicans* under stress conditions [57], we assessed the ammonia concentrations of the supernatants using a commercially available colorimetric assay that specifically detects ammonia [58]. We observed higher ammonia concentrations in supernatants from *C. auris* strains that melanized and had higher pH (Figure 9E,F).

To determine if the melanin structures that form in the cell-free supernatant from the cultures had similar ultrastructural characteristics as autopolymerized L-DOPA, we collected the oxidized L-DOPA from the supernatants of CDC 381, CDC 387, CDC 388, and minimal media alone at their baseline pH values (~5.75, 6.5, 6.5, and 5.5, respectively) and at pH 7. We imaged these melanized particles using negative staining TEM. Interestingly, we found that the melanized particles from CDC 387 and CDC 388 had structural differences compared with the oxidized L-DOPA in the minimal media at pH 7 (Figure 9G). The pigment particles collected from the cell-free supernatant incubated with L-DOPA were similar to those collected from melanized cultures, whereas the minimal media with auto-oxidized L-DOPA alone had smaller irregular particles and rougher clumps of electron-dense material (Figure 9G). This suggested that there is some component in the supernatant that may help in structuring the auto-polymerized L-DOPA, by acting as a scaffold, allowing it to form spherical and linear structures like beads on a string. Melanin is known to bind strongly to carbohydrates, lipids, and proteins, which could be such factors [19,24].

### 3.13. Melanin Protects C. auris from Oxidative Damage

*C. auris* cultures incubated, or not, with L-DOPA for 7 days were treated with hydrogen peroxide (H_2_O_2_), and yeast viability was evaluated (Figure 10A). The incubation with H_2_O_2_ reduced the viability of all evaluated strains of *C. auris*. The melanin-producing strains (CDC 387 and CDC 388) incubated with L-DOPA were partially protected from the H_2_O_2_, when compared to cells from the same strain grown in the absence of L-DOPA. Strain CDC 381 showed similar viability in the presence and in the absence of L-DOPA, indicating that melanin protects *C. auris* yeast cells against oxidative attack by H_2_O_2_.

### 3.14. Melanin Does Not Protect C. auris against Macrophage Killing

Melanized and control *C. auris* yeast cells were incubated with murine bone marrow-derived macrophages (BMDM) for 2 and 24 h and yeast killing was assessed. Under these conditions melanin did not confer protection to the yeast cells against macrophage killing (Figure 10B). It was interesting to observe that whereas the killing after 2 h was similar among the strains, after 24 h CDC 381 was shown to be partially resistant against macrophage killing, as opposed to the susceptible phenotype exhibited by strains CDC 387 and CDC 388.

### 3.15. Melanin Does Not Affect C. auris Virulence during Galleria Mellonella In Vivo Infection

Larvae of *G. mellonella* were infected with yeast cells from melanized and control *C. auris* and survival was evaluated. As seen for the killing experiment with mouse macrophages, the presence of melanin did not impact the survival of *Galleria mellonella* larvae (Figure 10C), suggesting that melanization of *C. auris* prior to infection did not confer protection to the fungus in this invertebrate model. We were unable to evaluate the production of de novo fungal melanin in the larvae during infection: *G. mellonella* produce catecholamine-derived melanin pigment as a key part of their immune response to fungi [59,60,61]. *C. auris,* much like other *Candida* spp., induces a robust melanization response in *G. mellonella* [62,63,64]. The insect melanin surrounding the fungal cells would be difficult to differentiate and separate from the fungi-produced melanin surrounding the fungal cells.

## 4. Discussion

In this work, we have investigated the ability of *C. auris* to produce melanin, a multifunctional pigment that is found across all biological kingdoms and contributes to the virulence of numerous pathogenic fungal species in mammals and plants. Some of the functions of melanin include the ability to neutralize reactive oxygen species during immune activation, inactivate antimicrobial peptides, and inactivate antifungal drugs [12,13,65]. Due to these immunity- and therapeutic-evasive properties, our findings that *C. auris* can melanize may have clinical relevance, particularly in regard to our understanding of how *C. auris* fungal infections are treated and considering this organism’s exceptionally high resistance to most common anti-fungal therapeutics. We evaluated whether *C. auris* can produce melanin, which strains produced the pigment, what substrates can be used for this melanization, where the melanin was localized, and the mechanism by which the melanin was produced.

### 4.1. Characterization of Melanin Production and Localization

We evaluated the ability of 18 *C. auris* strains to melanize when grown in the presence of L-DOPA, a commonly used substrate in fungal melanin research, and a common substrate for fungi that do not endogenously produce melanin. We found that most but not all of the strains melanized, which manifested by darkening of cultures, and that the ability to melanize was enhanced at higher temperatures (37 °C compared to 30 °C). We had anticipated that melanization would occur similarly in all strains, so it was surprising that only some strains had the ability to melanize. Further, based on our understanding of fungal melanization, we expected melanization to occur more at 30 °C rather than 37 °C, as it does in other fungi such as *Cryptococcus neoformans* [66]. This trend also has relevancy for pathogenesis in humans, as human body temperature is 37 °C and thus more conducive to ideal *C. auris* melanization conditions. Interestingly, the pattern of melanization correlated with the clade to which the strain belonged: *C. auris* strains from Clades I and IV, which are associated with South Asia and South America, respectively, are the melanizing strains, while the non-melanizing strains are from Clades II and III, which are typically associated with East Asia and South Africa, respectively. This demarcation could point to some lineage-specific genetic or epigenetic differences that are responsible for the melanization phenotype or are regulating the melanization process. Another aspect that differed from the pigmentation observed in other fungi was the impact of cell density on melanization by *C. auris* [67]. We observed that melanization in *C. auris* required high cell density (10^7^ cells/mL). However, at an intermediate cell density (10^6^ cells/mL) only some strains of *C. auris* (CDC 382, 385, 386, and 389) were able to produce melanin, and in a delayed fashion. This observation could indicate the need of a quorum sensing molecule to trigger effective melanization in *C. auris*.

Fungi have been shown to produce DHN-melanin, DOPA-melanin, and pyomelanin. DHN-melanin is produced from endogenous precursors formed through the polyketide synthesis pathway and not through exogenously added precursors [12]. Thus, we tested the production of melanin following addition of various DOPA-melanin and pyomelanin substrates. We found that *C. auris* produces melanins consistent with typical DOPA-melanins using substrates such as L-DOPA, D-DOPA, Methyl-DOPA, dopamine, epinephrine, and norepinephrine. *C. auris* was unable to produce melanin from L-Tyrosine, indicating there is no tyrosinase present to convert L-Tyrosine into L-DOPA and downstream melanin intermediates. Neither are there enzymes of the Tyrosine Degradation Pathway that would be needed for pyomelanin production, an observation supported by the inability to convert 4-HPP into pyomelanin pigment. The *C. auris* genome has hypothetical genes related to tyrosinase, as annotated by FungiDB, the online bioinformatics database for fungi, but these genes appear more related to ergosterol synthesis or magnesium/zinc ion transport. Similarly, *C. auris* was unable to form pyomelanin from homogentisic acid (HGA) and was unable to oxidize ABTS into its blue form. Laccases, such as those in *C. neoformans*, are able to produce a brown pyomelanin-like pigment from HGA and convert colorless ABTS into its blue oxidized form [42,68]. These results in *C. auris* suggest that a laccase does not exist as it does in *C. neoformans,* or at least none with as wide of a substrate range. Melanization in other DOPA-melanin producing fungi is often catalyzed by laccases [69] or multicopper ferroxidases [11]. Current annotation of the *C. auris* genome indicates there are hypothetical laccases, oxidoreductase, and ferroxidases (Ontology Groups OG6_100257 and OG6_100380); however, these appear to be related to metal ion homeostasis and part of the peroxisomal membrane. Since we do not know the function or expression pattern of these hypothetical genes, we cannot rule out that their products are capable of oxidizing catecholamines under certain conditions. However, our current body of evidence points to a non-laccase, non-tyrosinase, and general non-enzymatic mechanism as the responsible melanin-producing factor.

We investigated the cellular localization of the melanin in three selected *C. auris* strains: CDC 381 which is a non-melanizing strain, CDC 387 which is a strongly melanizing strain, and CDC 388 which is a moderately melanizing strain that produces pigment primarily at 37 °C. For some experiments, we included the MMC1 strain that, like CDC 388, is a moderately melanizing strain. Using light microscopy, we observed numerous dark aggregates on the outside of the melanized CDC 388, MMC1, and particularly CDC 387 cells, which tended to have a fluffy appearance and appeared in between areas where cells were clumped together. In addition, the cells themselves appeared darker, and some had large intracellular melanized spots which would appear to correspond to a vacuole. Based on our evidence that melanization occurs extracellularly due to the alkalinization of the supernatant, we believe these dark spots within the vacuole are due to L-DOPA autoxidation within the vacuole following storage under nutrient-deprived conditions. In the CDC 381 strain, these features were minimal to non-existent. To gain further understanding of the melanin on an ultrastructural level, we used TEM and SEM. In essence, we found that there were electron-dense spherical structures on the surface of the melanized *C. auris*, which were bound to or associated with polysaccharides in the outermost layer of the cell wall, likely β-glucans or mannoproteins, as indicated by the lack of melanin adhesion following the inhibition of β-glucan synthesis with caspofungin. These structures were only present in melanized cells, and they were consistent with previously reported melanin granules [9,19]. Interestingly, these structures were not reliably seen intracellularly, and they were significantly smaller than the extracellular melanin structures seen during *C. albicans* melanization [9]. Using SEM, we also saw these spherical particles on the surface of the melanized *C. auris* strains, especially the CDC 387 strains where the surface was heavily decorated in these structures. The CDC 387 cells had a distinctive phenotype featuring many multivesicular bodies (MVBs) within the cell and merging with the plasma membrane, resulting in secretion of extracellular vesicles. Extracellular vesicles contain protein, lipid, polysaccharide, and nucleic acid cargo that could be important for extracellular functions; secretion could be associated with *C. auris* resistance to amphotericin B, adhesion to epithelial cells, and survival within macrophages [70,71].

We noticed that the supernatants of the melanized cultures were quite dark, particularly for the CDC 388 and MMC1 strains, which indicated there was secreted melanin. We also made this observation on solid agar, where all the melanizing strains had a halo of melanin around the colonies and the non-melanizing strains did not. The strongly melanizing strains CDC 385, CDC 386, CDC 387, and CDC 389 had the darkest halos of melanin surrounding the colonies. The fungal colonies themselves remained white with no change in pigmentation after 7 days of growth. The notion that the melanin is extracellular and secreted is supported by TEM and SEM results that show melanized structures on the surface of the cells. Using negative staining TEM of the isolated melanin from the supernatant, we saw melanin granules similar to those seen in TEM of whole cells, where the CDC 387 granules were the largest, followed by MMC1 and CDC 388. The CDC 381 granules were the smallest and fewest. When we added the melanized supernatant to non-melanized cells, including to CDC 381, we found that the melanin adhered to the cells and took on the appearance of a melanized cell pellet. These findings indicated that the extracellular melanin could adhere to the outside of the cell wall, suggesting a scheme whereby melanization occurred externally and aggregates of the polymer adhered to the cell wall.

### 4.2. Mechanism of Melanin Production

Although our data point to a role for the cell wall in melanin adherence, they also show that the ability of the cell wall to host adhering melanin is unrelated to melanin production itself, which appears to be an exclusively extracellular process. First, when L-DOPA was added to the cell-free supernatant of the cultures, we saw that the supernatants from CDC 388 and CDC 387 had melanizing activity whereas the supernatant from CDC 381 did not, indicating that the component responsible for melanization was extracellular. Second, melanin isolated from the culture supernatant of the melanizing CDC 388 strain can adhere to the cell wall of CDC 381 strain cells, which themselves are unable to produce melanin. Third, our ssNMR studies suggest that cell wall melanin adherence is related to the relative proportion of polysaccharides that are found in the cell wall. CDC 387 strain cells, which display robust cell-wall melanin adhesion, were found to have a greater proportion of polysaccharides in comparison to CDC 388 or 381 cells, which were determined to have a similar relative polysaccharide content and also display a similar degree of cell-wall melanin adhesion. Taken together, these findings demonstrate that melanin production and cell-wall melanin adherence are two unrelated processes and that the lack of melanization activity of the CDC 381 strain is not due to an inherent deficiency of the cell wall.

To determine the extracellular factor responsible for melanization, we performed a series of assays on the cell-free supernatant. First, we determined that the melanization factor was stable at 100 °C, was resistant to proteolytic degradation by trypsin and Proteinase K and was smaller than 3 kDa. These observations together indicated that the melanization component was not a protein, but could potentially be a small molecule. Further, the melanizing ability of the supernatant is not enriched in the extracellular vesicles collected through ultracentrifugation, nor is it protected by EVs. We cannot exclude the possibility that EVs are used as scaffolding for the melanization, as previously found in *C. neoformans* [72]. However, melanization activity is not lost after treatment with methanol, urea, or SDS detergents.

Interestingly, we found that the ability of a strain to melanize correlated directly with the pH of the supernatant within minimal media, where non-melanizing strains had a supernatant pH of ~5.75 and melanizing strains had a pH ~6.5. This strong correlation also corresponds to the steep increase in autoxidation of L-DOPA between those pH ranges. We additionally found that artificially adjusting the pH of the supernatant to pH 7 caused all of the supernatants to melanize, whereas adjusting the pH to 5.5 prevented all the supernatants from melanizing, even those from the melanizing strains. This may explain why melanization was enhanced in all supernatant conditions tested following the 6 M urea treatment, which likely neutralizes or alkalinizes the pH of the supernatant. These data suggest that the melanization trends reflect the ability of some strains to alter the acidity of the surrounding environment more than others. L-DOPA autoxidation is highly dependent on pH [17,18]. L-DOPA has an isoelectric point of pH 6.0, meaning that at pH 6 or above, the L-DOPA is more likely to have a deprotonated amine group, which results in more energetically favorable oxidation, cyclization, and dopachrome formation [18]. We found strong evidence that this ability to neutralize the media was due to the production of ammonia by some of the strains. Fungi including *C. albicans* and *C. neoformans* are known to produce ammonia to boost their persistence under pH stress [57,73,74]. *C. auris* has been found in environmental reservoirs within marine saltwater marshes [2]. Microbes found in marine wetland environmental niches are faced with alkaline stress and, as a result, may be naturally more alkaliphilic [75].

Whereas the autoxidation of catecholamines could occur in the absence of any fungus, we found that the oxidized L-DOPA from the supernatant of the *C. auris* strains had a different ultrastructure than the autoxidized L-DOPA from the minimal media alone at a neutral pH. This indicated that there are some properties of the supernatant that encourage the autoxidized L-DOPA to organize into specific spherical structures. Based on previous understanding of how melanin interacts with other biological molecules, we hypothesize this melanin scaffold to be polysaccharide, protein aggregates, or perhaps even lipids. The structure of the melanin polymer around this scaffolding could change its biophysical properties. This hypothesis could help to explain why there is significantly more melanin associated with the CDC 387 strain—the strain changes the pH of the supernatant the same as the other strains, but the electron micrographs also show a large number of MVBs and organelles fusing with the plasma membrane of these cells, which would indicate a relative hypersecretion of polysaccharides, proteins, and extracellular vesicles that can subsequently serve as a scaffold for the melanization. Interestingly, the supernatant of the non-melanized CDC 387 strain has a yellow tint to it, which indicates that some molecules are being released in higher amounts compared with the CDC 381 and CDC 388 strains. This pH-based mechanism of fungal melanization differs greatly from other ways in which fungi such as *C. albicans* produce melanin [9,10,11]. In *C. albicans,* melanization is mediated by ferroxidases, a class of enzymes that is genetically similar to the melanin-producing enzyme laccase in *C. neoformans.* The knockout of some ferroxidase genes (*FET* genes) results in a loss of melanization phenotype.

### 4.3. Implications of C. auris Melanization

We found that the *C. auris* strains, generally, were remarkably hydrophobic, with the cell surface hydrophobicity (CSH) of most strains in the 90–100% range. Comparatively, the CSH of clinical isolates of *C. albicans* strains ranges from about 2 to 41% [47]. In *C. albicans*, CSH is reportedly associated with increased adhesion to epithelial cells, biofilm formation, resistance to neutrophil-mediated killing, and increased overall virulence [48,76,77,78]. The current findings have implications for how *C. auris* interacts with hydrophobic surfaces, and how hydrophobicity may contribute to durable biofilm formations within hospital environments and medical equipment. We do not find an association between CSH and ability of the strain to melanize. Since melanin is a hydrophobic polymer, we anticipated that melanization of cultures would enhance CSH. Surprisingly, we found that growth in minimal media with L-DOPA reduced the hydrophobicity of the non-melanizing strain CDC 381 and the strong melanizing strain CDC 387, while the CSH of CDC 388 and MMC1 were virtually unchanged. We hypothesize that the hydrophobic melanin is comparatively less hydrophobic than the hydrophobicity-mediators on the surface of the *C. auris* cells, so that when the melanin binds to the cell wall, it results in a relative decrease of CSH. In *C. albicans*, CSH is believed to be mediated by exposure of hydrophobic cell wall mannoproteins [49,79], which could correspond to areas in which melanin is deposited in *C. auris* [80].

One notable observed feature of the heavily melanized *C. auris* CDC 387 strain is that it tended to form large aggregates of cells, which were melded together by extracellular melanin, as indicated by light and electron microscopy. This aggregation is notable, as aggregation is a known physical property in some strains of *C. auris*, outside of the context of melanin. The aggregate phenotype has been hypothesized to function as a way for the fungus to remain within tissues and evade immune clearance, and it may play a role in biofilm formation and maintenance [81,82,83]. If melanization is inducing a sort of aggregative phenotype as our data suggest, this could have similar implications in understanding the role of *C. auris* melanin in pathogenesis.

The antioxidant properties of melanin are partially responsible for its protective properties [84]. Melanized *C. auris* yeast cells were partially protected against hydrogen peroxide, and this protection was stronger in the strain where melanization is more efficient (CDC 387). This observation is in accordance with the expected properties of melanin pigments [12,65]. However, melanin did not confer any protection to *C. auris* in an in vitro challenge with BMDM. The in vitro killing of *C. auris* by BMDM was tested at distinct times and, under the evaluated conditions, melanin played no role on protecting the fungus from the phagocytes. The previously described degree of protection from killing of pathogenic fungi conferred by melanin varies considerably (18% to >36%) [85], so it is possible that the pattern observed for *C. auris* is similar to *C. albicans*, for which there is no clear association between ability to produce melanin and virulence in mice [9]. Supporting the in vitro findings, in vivo experiments using the invertebrate *G. mellonella* showed that melanin produced by *C. auris* might not protect the fungus against innate immunity, but we cannot rule out a potential action of melanin to interfere in processes that lead to acquired immunity.

## 5. Conclusions

In summary, we found that Clade I, IV, and V strains of *C. auris* grown with L-DOPA and catecholamine substrates produce melanin extracellularly by alkalinizing the media with ammonia, which promotes non-enzymatic catecholamine oxidation. This extracellular melanin aggregates and binds to the outside of the *C. auris* cell wall (summarized in Figure 11). We did not find evidence that the melanin formed by *C. auris* interferes with the effector mechanisms of innate immune cells, nor does it exhibit an active melanization process. However, our findings leave open the possibility that extracellular alkalinization could be a new mechanism by which fungi can drive the production of melanin in the environment. Not much is known about the environmental niche of *C. auris;* however, the first environmental isolates have been uncovered and described in the neutral pH marine marshlands off the Andaman Islands in India, and on the surface of non-freshly picked apples in Northern India, all of which have been members of the South Asian Clade I [2,86]. It is possible that melanization may play a role in the environmental survival of *C. auris*, where the fungus is likely to encounter oxidative stressors. Ammonia is produced by other fungal species and namely in *C. albicans*: ammonia de-acidifies the extracellular space, is produced during nutrient deprivation, and auto-induces morphologic and metabolic changes [57,74]. The production of ammonia and alkalinization of the extracellular milieu in *C. auris* might have similar roles. *C. auris* extracellular alkalinization might be the result of adaptations to acidic or stressful environments and aid in fungal survival uncoupled from the melanization process. With future investigations uncovering global environmental niches of *C. auris* from all strains, better insight can be gained into the biological significance of our findings of clade-specific *C. auris* ammonia production, alkalinization, and melanization.

## Figures and Tables

**Figure 1 jof-08-01068-f001:**
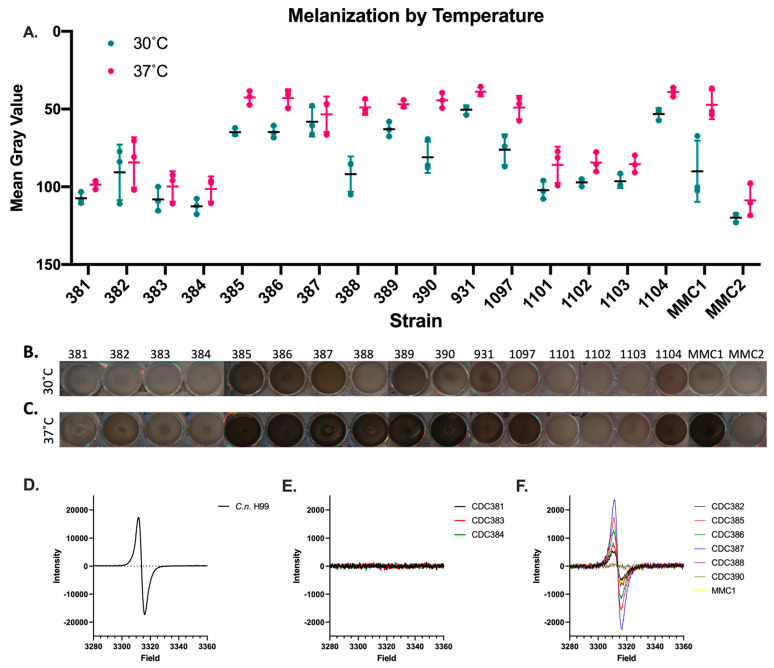
*Candida auris* produces melanin pigment in a strain− and temperature−dependent manner. Melanization occurs in only some of the *C. auris* strains and the extent of melanization is variable. This melanization can be quantified (**A**). Melanization occurs less at 30 °C (**B**) than at 37 °C (**C**), particularly for the melanizing strains CDC 388, CDC 390, and MMC1. Experiments were performed in three biological replicates. Panels (**B**,**C**) are representative images. (**D**–**F**) Electron paramagnetic resonance (EPR) of melanin isolated from *Cryptococcus neoformans* (**D**), non−melanizing *C. auris* strains (**E**) and melanizing *C. auris* strains (**F**) grown in minimal media with L–DOPA. Error bars represent standard deviations from three independent experiments.

**Figure 2 jof-08-01068-f002:**
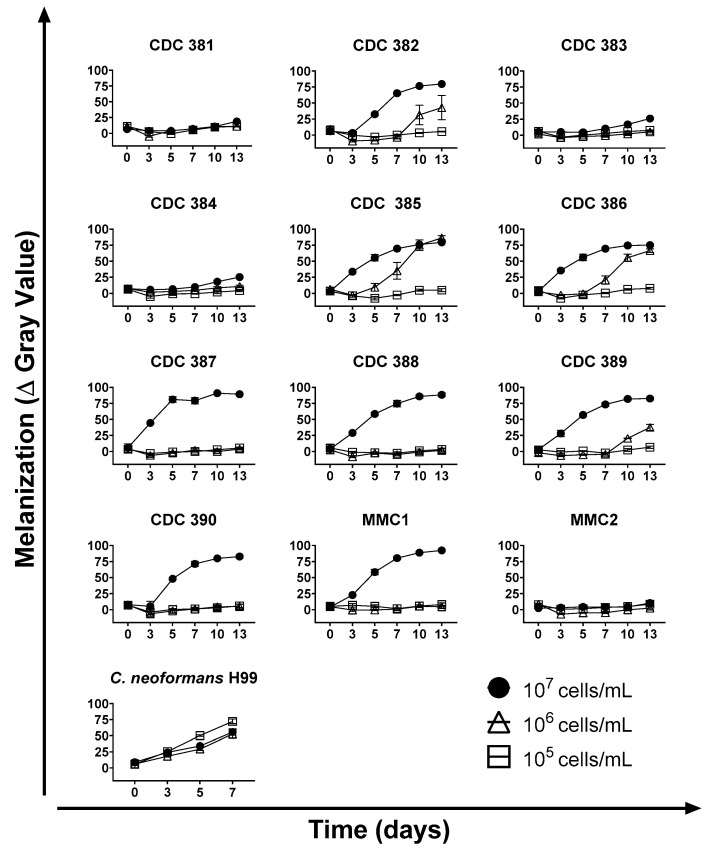
Cell density and time course. *C. auris* yeast cells were incubated at increasing cell densities in minimal media in the presence or absence of L–DOPA. Melanin formation was monitored through the course of 13 days and quantified using ImageJ. The graphs show means and standard deviation from 3 independent experiments.

**Figure 3 jof-08-01068-f003:**
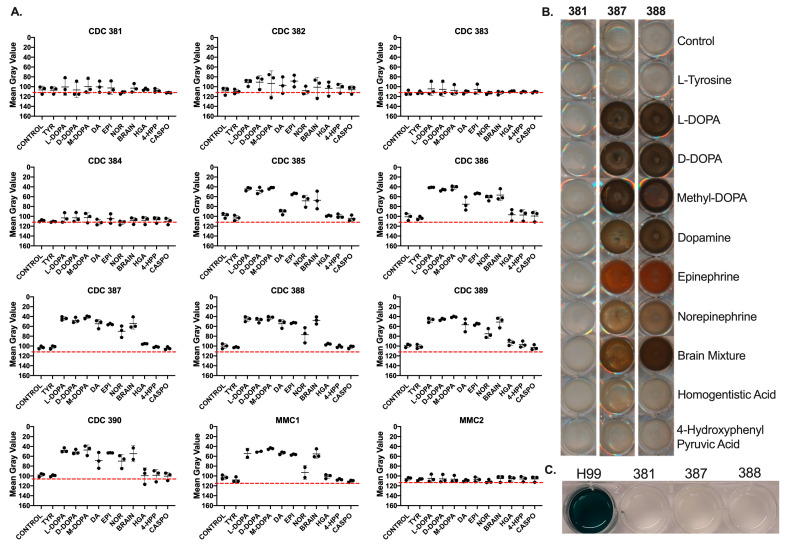
Melanization by substrate. (**A**) *C. auris* melanizing strains use catecholamines such as L–DOPA, D–DOPA, Methyl–DOPA, dopamine, epinephrine, and norepinephrine as substrates for melanization. Overall, there are few differences between the melanizing strains’ abilities to oxidize these substrates. (**B**) The color of the pigments formed was typically brown-black, with the exception of the orange produced with epinephrine. Red dotted lines represent the baseline mean gray value of the background. Values above the line (closer to 0) signify wells darker than the background. (**C**) Strains of *C. auris*, CDC 381, 387, and 388 do not have laccase activity when grown with the laccase-specific substrate ABTS, whereas the positive control—the H99 strain of *Cryptococcus neoformans*—does have laccase activity, as indicated by the blue color formed in culture. Data represent biological triplicates, with Panels (**B**,**C**) showing representative images of the melanized cultures. Abbreviations: L-Tyrosine (TYR), Methyl–DOPA (M–DOPA), dopamine (DA), epinephrine (EPI), norepinephrine (NOR), Brain mixture (BRAIN), homogentisic acid (HGA), 4-hydroxyphenylpyruvic acid (4-HPP), Caspofungin (CASPO). Error bars indicate standard deviation from the mean.

**Figure 4 jof-08-01068-f004:**
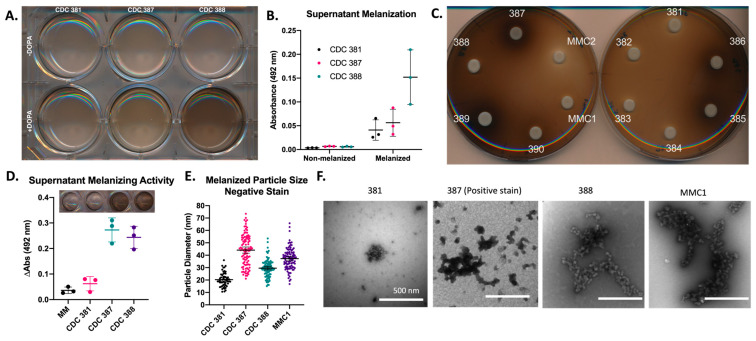
Melanization primarily occurs extracellularly in *C. auris*. (**A**,**B**) Melanized cultures have darker supernatants than their non-melanized counterparts, particularly the supernatant from the CDC 388 strain. (**C**) When grown on solid agar, all melanizing strains have a dark halo surrounding their colony, with darker color around strong melanizing strains. The colonies of the strains themselves are notably white. (**D**) The cell-free supernatants from CDC 387 and CDC 388 (melanin-capable strains) have the ability to melanize, indicating a melanization factor is present. (**E**,**F**) Cultures grown with L-DOPA have melanin particles in the supernatant that can be collected and visualized via electron microscopy. The strongly melanizing CDC 387 strain has the largest population of these particles, the moderate melanizing strains CDC 388 and MMC1 have moderately sized populations of these particles, and the weak melanizing strain CDC 381 has the smallest population. All experiments were done in biological triplicate. Panels (**A**,**C**,**F**) are representative images. Scale Bar indicates 500 nm. Error bars indicate standard deviation.

**Figure 5 jof-08-01068-f005:**
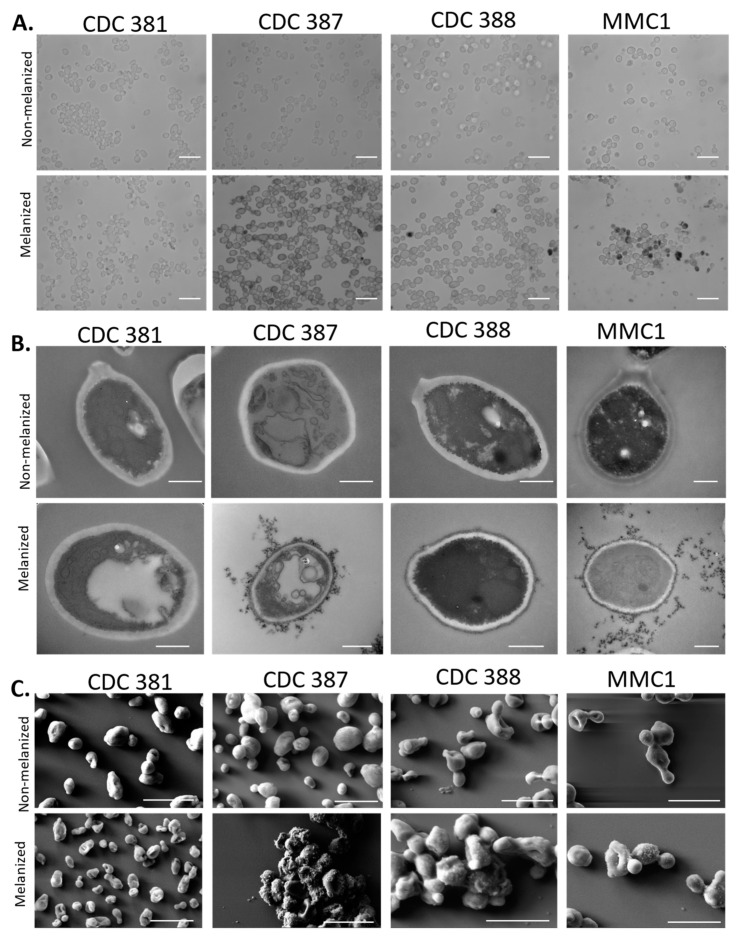
Melanin is primarily deposited on the cell wall surface. (**A**) Light Microscopy shows that the 387, 388, and MMC1 cells have darker color when grown in minimal media with L-DOPA, while there is little color change in the CDC 381 cell color. Some of the pigmented cells have dark pigmentation within the cell. CDC 387 in particular appears to have large pigmented extracellular structures. Scale bars represent 10 µm. Images representative of three biological replicates. (**B**) Transmission electron microscopy shows electron-dense particles on the outside of only the 387, 388, and MMC1 cells grown in the presence of L–DOPA, which is indicative of melanin pigment. These dark particles are found on the external cell wall and in the extracellular space; they are especially visible in the CDC 387 and MMC1 micrographs. Scale bars represent 500 nm. (**C**) Similar external granule structures on the surface of CDC 387, 388, and MMC1 cells grown with L–DOPA can be seen in scanning electron micrographs. Scale bars represent 5 µm.

**Figure 6 jof-08-01068-f006:**
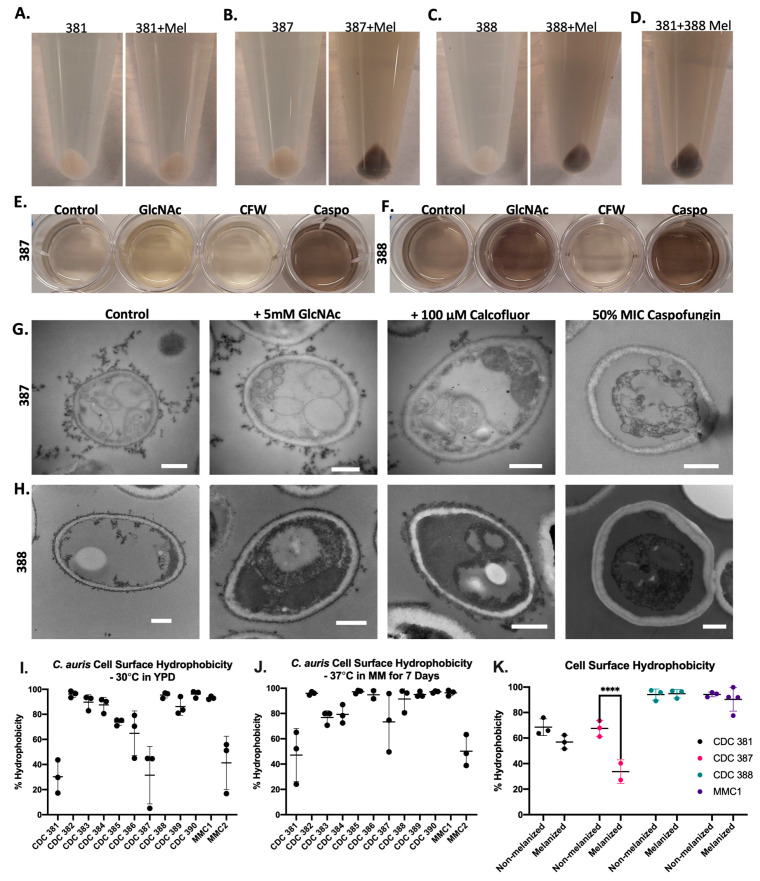
Extracellular melanin adheres to cell wall and is blocked following β−Glucan inhibition. (**A**–**C**) Adding non−melanized cells to the melanized supernatant from its corresponding strain resulted in pigmented cells after 3 h of incubation. (**D**) The melanized supernatant from the CDC 388 strain causes the non-melanizing CDC 381 strain to become pigmented. Adding the building block for chitin, N-acetylglucosamine (GlcNAc), to the CDC 387 (**E**) and CDC 388 (**F**) strains resulted in increased supernatant melanization, and blocking chitin synthesis with CFW resulted in less melanin in supernatant but also caused melanin−coated CFW crystals on the fungal surface. Blocking β−Glucan synthesis with caspofungin resulted in increased supernatant melanization in both strains. (**G**,**H**) Electron microscopy revealed no major differences in melanin adherence to the cell wall in either the CDC 387 (**G**) and CDC 388 (**H**) strains treated with GlcNAc. There were clear electron−dense crystal structures on the cell walls of CFW conditions, and the caspofungin-treated conditions had little to no melanin on the cell surface. While caspofungin−treated cells grew in culture, they look atypical in TEM with a condensed cytoplasm. All scale bars represent 500 nm. (**I**,**J**) There is no association between the hydrophobicity of a *C. auris* strain and its ability to melanize. (**K**) Although the melanization of certain strains, namely CDC 387, results in a significant reduction of CSH compared to its non-melanized control, no statistically significant reduction of CSH occurs in CDC 381, 388, or MMC1 strains. All experiments are representative of two to three biological replicates. Statistical differences were tested with a Two−Way ANOVA with multiple comparisons using Prism GraphPad, with *p* < 0.0001 = ****. Error bars indicate standard deviation.

**Figure 7 jof-08-01068-f007:**
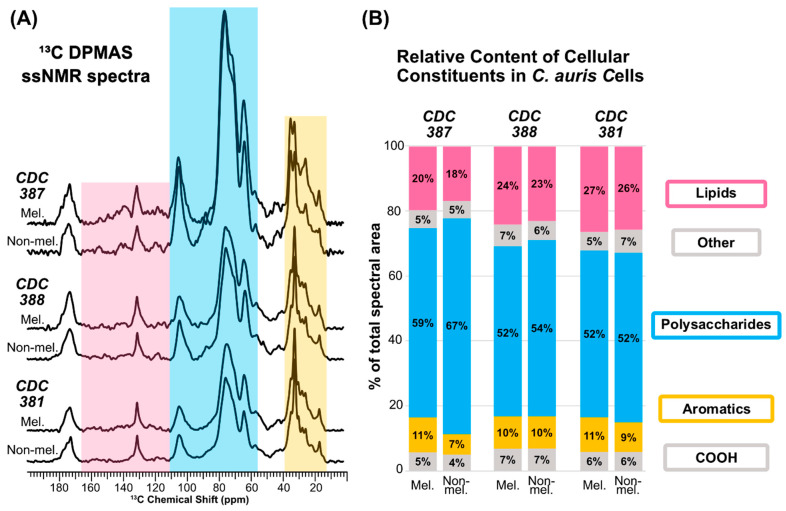
Melanin adherence is correlated with a higher relative cell-wall polysaccharide content. (**A**) Quantitatively reliable direct-polarization solid-state ^13^C NMR spectra of intact heat-killed *C. auris* cells from the melanizing strains CDC 387 (top) and CDC 388 (middle), and the non-melanizing strain CDC 381 (bottom) grown with and without L-DOPA. The spectral regions are shaded according to the predominant type of molecular structure: yellow (12–37 ppm), aliphatic carbons of lipids; blue (54–110 ppm), polysaccharide ring carbons; pink (110–166 ppm), aromatic carbons of protein side chains, nucleic acids, and melanin pigments plus C=C from lipids (130 ppm). The signals in the unshaded regions arise from carbon types found in many different molecules rather than a single group of cellular constituents: 37–54 ppm, tertiary carbons, amino acid carbons, and methoxy carbons; 166–182 ppm, carbonyl carbons within various carboxyl groups. (**B**) Bar graph displaying estimates of the relative content of polysaccharides and other groups of cellular constituents with respect to their respective total content in intact heat-killed *C. auris* cells. The data represent the mean estimates from three independent sets of quantitative measurements; the standard deviation from the mean was less than 0.04% for these measurements.

**Figure 8 jof-08-01068-f008:**
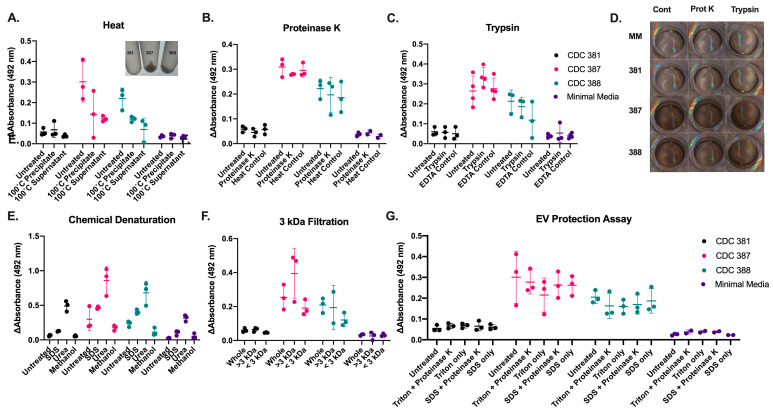
*C. auris* melanization factor is resistant to denaturing conditions. (**A**) Heating the supernatant to 100 °C slightly reduces the melanizing ability of the supernatant but does not completely abolish melanizing ability of the supernatant or the precipitate that forms following boiling (inset). Adding proteinase K (**B**) or Trypsin (**C**) does not reduce the melanizing ability of the supernatant. (**D**) Images of the oxidation after 72 h within the samples treated with proteinase K and trypsin (**E**). Similarly, 33% methanol and 1% SDS did not reduce melanization activity, whereas 6 M urea uniformly enhanced DOPA oxidation in all groups. (**F**) Filtration through a 3 kDa filter indicates that the melanizing factor is smaller than 3 kDa. (**G**) The melanizing factor is not protected from proteolytic degradation by being shielded within extracellular vesicles (EV), as addition of detergent to disturb EVs does not render the melanization factor more susceptible to proteolytic degradation. Error bars indicate standard deviation.

**Figure 9 jof-08-01068-f009:**
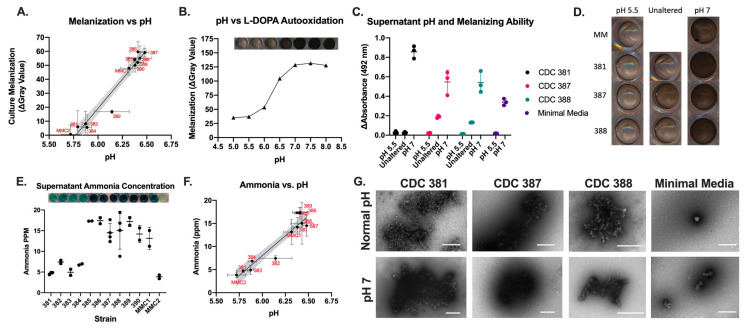
Supernatant pH correlates to melanization ability. (**A**) The pH of the supernatant from the *C. auris* cultures corresponds to the ability of that strain to melanize, with two distinct clusters that correspond to melanizing versus non-melanizing strains. (**B**) The correlation between pH and melanization from the *C. auris* strains corresponds to a similar increase in autoxidation of L–DOPA in the same range of pH. (**C**) Reducing the pH of the supernatants to 5.5 abolished melanization ability in the CDC 387 and 388 strains, while raising the pH of the supernatants to 7 enhanced melanization in all the supernatants, including CDC 381. (**D**) Images of the L–DOPA oxidation in the supernatant following pH alteration. (**E**,**F**) *C. auris* strains that melanize are associated with increased ammonia concentration in the supernatant, with a concentration of approximately 16 ppm, whereas non-melanizing strains have ammonia concentrations between 8 and 5 ppm. (**G**) Aggregates from pH 7 melanized cell-free supernatant from CDC 387 and CDC 388 look similar to those produced in cultures (Figure 4F), whereas the pigment collected from the pH 7 minimal media looks more amorphous and lacks structure. The Scale Bar represents 500 nm. Error bars indicate standard deviation from the mean value.

**Figure 10 jof-08-01068-f010:**
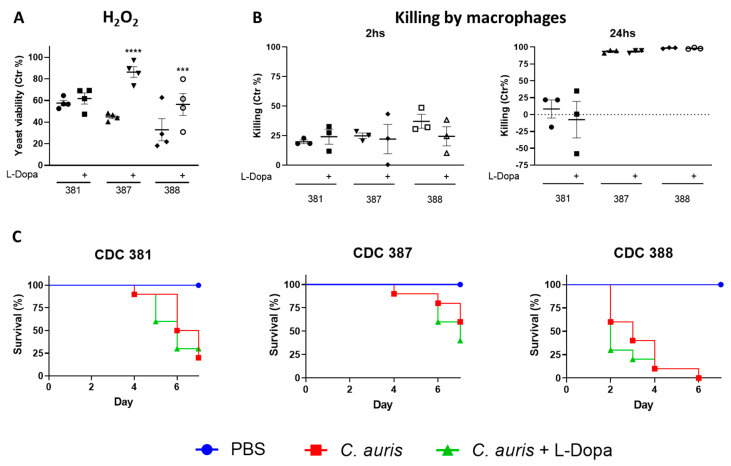
Melanin from *C. auris* during interaction with the host. (**A**) Melanized and control yeast cells were incubated with, or without, 5 mM hydrogen peroxide (H_2_O_2_) for 3 h at 37 °C. Yeast viability was addressed by CFU counting, and values of H_2_O_2_−treated *C. auris* were expressed as a percentage of cells in the absence of H_2_O_2_. Each symbol in the graph represents one experiment performed in triplicate. Data was treated by One-way ANOVA followed by Sidak’s multiple comparison test. *** represents *p* = 0.0003, **** represents *p* < 0.0001. (**B**) Bone marrow-derived macrophages (BMDM) were challenged with melanized or control *C. auris* yeast cells for 2 or 24 h. After macrophage lysis, fungal viability was addressed by CFU counting. The killing percentage was expressed as the amount of viable yeast in the presence of macrophages, divided by the amount of yeast in the absence of macrophages for each strain and condition. Each symbol represents one experiment performed in triplicate. (**C**) Melanized and control *C. auris* yeast cells were inoculated into the haemocoel of *G. mellonella* larvae (*n* = 10/group). The number of living larvae was monitored for 7 days after inoculation.

**Figure 11 jof-08-01068-f011:**
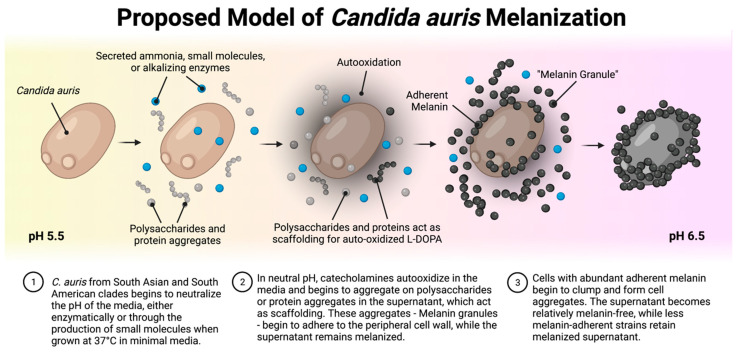
Proposed model for *C. auris* melanization. *C. auris* from Clades I and IV can neutralize the pH of their surrounding environment by secreting either ammonia, an alkalinizing enzyme, or some other pH modulator. The neutralized pH encourages oxidation of catecholamines that are present, which begin to form a pigment. The oxidized catecholamines may aggregate on secreted proteins, polysaccharides, or lipids to form melanin granules. These melanin granules are deposited on the surface of the fungi. In cases of high degrees of melanization, nearly all of the extracellular melanin ends up adhering to the surface of the fungus, leaving a relatively light-colored supernatant.

**Table 1 jof-08-01068-t001:** *C. auris* strains, clades, and their ability to melanize.

Strain	Alias	Clade	Melanin
CDC 381	B11220	Clade II (East Asian)	No
CDC 382	B11109	Clade I (South Asian)	Partially
CDC 383	B11221	Clade III (African)	No
CDC 384	B11222	Clade III (African)	No
CDC 385	B11244	Clade IV (South American)	Yes
CDC 386	B11245	Clade IV (South American)	Yes
CDC 387	B8441	Clade I (South Asian)	Yes
CDC 388	B11098	Clade I (South Asian)	Yes
CDC 389	B11203	Clade I (South Asian)	Yes
CDC 390	B11205	Clade I (South Asian)	Yes
CDC 931	B11243	Clade IV (South American)	Yes
CDC 1097	IFRC2087	Clade V (Iranian)	Yes
CDC 1101	B18678	Clade II (East Asian)	No
CDC 1102	B17835	Clade III (African)	No
CDC 1103	B18683	Clade III (African)	No
CDC 1104	B18017	Clade IV (South American)	Yes
MMC1	N/A	Clade I (South Asian)	Yes
MMC2	N/A	Unknown	No

## Data Availability

Data will be made publicly available through Figshare (https://figshare.com/projects/Melanization_of_Candida_auris_is_Associated_with_Alteration_of_Extracellular_pH/146688). Links to the underlying data can also be found in Appendix A.

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
