# Peer review of "Melanization of Candida auris Is Associated with Alteration of Extracellular pH"

_jof, 2022, doi:10.3390/jof8101068_

Round 1

Reviewer 1 Report

The authors provide extensive evidence for and characterization of melanin produced through alkalinization of the extracellular medium surrounding C. auris. The manuscript is well-written and a large amount of data is presented. A couple of questions/considerations for the authors:

1. Is there any correlation between strains that produced melanin and antifungal drug resistance patterns? This information could be added to Table 1.

2. EPR will be an unfamiliar technique for many readers of the JOF so a basic explanation of what it is measuring is needed earlier in the manuscript. An explanation appears in the Results section but EPR is introduced earlier in the Methods section so moving that explanation up to earlier in the manuscript would help readers.

3. The manuscript is quite long. It could be condensed by moving some information to the Supplemental section (e.g., Figure 7) and by condensing the introduction, results and discussion text.

Author Response

  1. Comment: The authors provide extensive evidence for and characterization of melanin produced through alkalinization of the extracellular medium surrounding C. auris. The manuscript is well-written and a large amount of data is presented. A couple of questions/considerations for the authors:

Response: we appreciate the reviewer’s comments, suggestions, and support for the manuscript.

  1. Comment: Is there any correlation between strains that produced melanin and antifungal drug resistance patterns? This information could be added to Table 1.

Response: We have attached a figure showing the antifungal susceptibility for these strains, as reported by the CDC and grouped by whether they are able to produce melanin or not. The strains capable of producing melanin do tend to have higher MICs compared to the strains unable to produce melanin. This is interesting, but we are not sure we can draw any conclusions from this correlation, since the melanizing and non-melanizing strains are from different clades, represented by many genetic differences and adaptations that can be unconnected from alkalinization and/or melanization.  We have placed it in the supplemental information as Figure S1 and have discussed the data briefly in the results section.

  1. Comment: EPR will be an unfamiliar technique for many readers of the JOFso a basic explanation of what it is measuring is needed earlier in the manuscript. An explanation appears in the Results section but EPR is introduced earlier in the Methods section so moving that explanation up to earlier in the manuscript would help readers.

Response: Thank you for pointing this out. We have added a few sentences to the EPR methods section like those in the results to help clarify the technique and its purpose for readers.

  1. Comment: The manuscript is quite long. It could be condensed by moving some information to the Supplemental section (e.g., Figure 7) and by condensing the introduction, results and discussion text.

Response: We would like to thank the reviewer for their feedback.  The reason the manuscript is long is because it presents an extensive body of data.  We considered the reviewer’s suggestion about Figure 7, but we feel the NMR-based correlation of melanin adhering strains with cell-wall polysaccharide content is essential to the story so we have retained this figure along with associated text within the body of the manuscript.  

Reviewer 2 Report

The authors have investigated the melanization properties of Candida auris strains in great detail. C. auris has emerged recently as a very hardy fungal pathogen. Therefore, the study is highly significant. Interesting findings are:

- Several strains or C. auris are capable of utilizing catecholamines such as L-DOPA and related compounds to produce melanin.

- Melanization occurs extracellularly and melanin particles can adhere to the cell walls of some C. auris strains.

- The “melanization factor” that leads to the autoxidation of L-DOPA in the culture medium does not appear to be a protein.

- Ammonia production and alkalization of the medium appear to contribute to melanin production.

- Melanin producing C. auris strains seem to be more resistant to oxidative stress (H2O2 experiment), however, they do not appear to have an advantage over non-melanizing strains when attacked by immune cells.

Overall, the experiments were well performed and described. The observation that urea treatment of supernatants increased the melanin-related absorbance at 492 nm for all tested melanizing strains (Fig. 8E), should have been followed up with an analysis of the melanin particle sizes. If urea denatures melanin particles, it could explain the apparent rise in absorbance.

Furthermore, while partially addressed, the oxidative potential of the C. auris strains could be further analyzed by testing if melanization in supernatants occurs in presence of reducing agents such as DTT or tris(carboxyethyl) phosphine, or by addition of antioxidants such as BHT. Perhaps a combination of alkalization and oxidation promotes melanin particle formation.

The Galleria mellonella experiments raise the question if infected larvae showed any signs of melanization at the fungus infected tissues.

Otherwise, the study is very detailed and provides novel important insight into the biochemical aspects of the C. auris pathogen.

Author Response

  1. Comment: The authors have investigated the melanization properties of Candida auris strains in great detail.  auris has emerged recently as a very hardy fungal pathogen. Therefore, the study is highly significant. Interesting findings are:
    • Several strains or  aurisare capable of utilizing catecholamines such as L-DOPA and related compounds to produce melanin.
    • Melanization occurs extracellularly and melanin particles can adhere to the cell walls of some C. auris strains.
    • The “melanization factor” that leads to the autoxidation of L-DOPA in the culture medium does not appear to be a protein.
    • Ammonia production and alkalization of the medium appear to contribute to melanin production.
    • Melanin producing  aurisstrains seem to be more resistant to oxidative stress (H2O2 experiment), however, they do not appear to have an advantage over non-melanizing strains when attacked by immune cells.

Response: We appreciate the reviewer’s summary of our work and their appreciation of our findings.

  1. Comment: Overall, the experiments were well performed and described. The observation that urea treatment of supernatants increased the melanin-related absorbance at 492 nm for all tested melanizing strains (Fig. 8E), should have been followed up with an analysis of the melanin particle sizes. If urea denatures melanin particles, it could explain the apparent rise in absorbance.

Response: This is a good point. Since the non-melanizing CDC 381 supernatant and minimal media-only control have similar increases in melanization following urea treatment, we believe this observation is due to the increased pH following addition of 6 M urea and/or the low concentrations of ammonia present in the urea as a breakdown product. We have added a sentence to briefly address this in the discussion.

  1. Comment: Furthermore, while partially addressed, the oxidative potential of the  auris strains could be further analyzed by testing if melanization in supernatants occurs in presence of reducing agents such as DTT or tris(carboxyethyl) phosphine, or by addition of antioxidants such as BHT. Perhaps a combination of alkalization and oxidation promotes melanin particle formation.

Response: This is a good point. We analyzed the effects of pH alterations on the ability for the strains to melanize, as pH is an important factor in controlling the speed of L-DOPA oxidation. However, in our studies to determine the mechanism by which C. auris melanization occurs, we did not investigate the effects that antioxidants/reducing agents had on our system as they are well known melanin inhibitors and would be likely to inhibit melanization regardless of the mechanism -- especially one that is non-enzymatically driven and based on the auto-oxidation of L-DOPA in the supernatant.

  1. Comment: The Galleria mellonellaexperiments raise the question if infected larvae showed any signs of melanization at the fungus infected tissues.

Response: This is another good point, but we did not investigate fungal melanization within the infected larvae. An important part of the insect immune response is the production of their own melanin, and Candida spp., in particular, elicit a strong melanization immune response. It would thus be difficult to separate the host-derived and the fungus-derived melanins, as they have similar physical properties and chemical composition and would be produced in the same space within the larvae. We have included a sentence in the results explaining why we did not look at in vivo fungal melanin production during infection.

  1. Comment: Otherwise, the study is very detailed and provides novel important insight into the biochemical aspects of the  auris pathogen.

Response: Thank you for the support of our work. We gratefully acknowledge that the reviewer’s suggestions have improved the manuscript.